# Development of flipped classroom module fcm for music theory instruction: An innovative approach to music education

Qisen Zhu[1], Khairul Azhar Jamaludin[1]*, Jinglong Li[2]

**1** Faculty of Education, University Kebangsaan Malaysia, Selangor, Malaysia, **2** Department of Industrial Design, Faculty of Design and Architecture, Universiti Putra Malaysia, Serdang, Malaysia

* khairuljamaludin@ukm.edu.my

## Abstract

Conventional lecture-based instruction in music theory often falls short in fostering deep engagement, critical thinking, and applied musicianship—particularly in educational contexts shaped by teacher-centered traditions. This study presents the design, implementation, and empirical evaluation of a Flipped Classroom Module (FCM) tailored to undergraduate music theory instruction in Chinese higher education. Grounded in a tripartite theoretical framework integrating Self-Directed Learning, Course Development Theory, and Collaborative Learning Theory. The module employs a three-phase instructional sequence—pre-class preparation, in-class collaboration, and post-class reflection—delivered through a customized learning management system. A quasi-experimental design (N = 60) compared the FCM group with a traditional instruction control group across both cognitive and practical learning domains. Validated pre- and post-tests assessed music-theoretical knowledge and applied skills, and statistical analyses (independent and paired-sample t-tests) revealed significant learning gains in the FCM group (p < .05). Beyond confirming the pedagogical efficacy of the flipped approach, the study demonstrates how culturally responsive instructional design can enhance accessibility, learner autonomy, and instructional coherence within constrained curricular environments. The findings contribute to ongoing international discourse on digital transformation in higher music education and offer a replicable framework for theory-informed, discipline-specific flipped pedagogy.

## 1. Introduction

Music theory constitutes a foundational component of undergraduate music education, providing the cognitive and structural framework for students to analyze, compose, and interpret music [1–3]. However, in Chinese higher education, music theory instruction remains predominantly lecture-based, often emphasizing rote memorization over

**Data availability statement:** All relevant data underlying the findings of this study are available in the Figshare repository at [https://figshare.com/s/53b3f260350779974fab].

**Funding:** The author(s) received no specific funding for this work.

**Competing interests:** No.

conceptual understanding and application [4]. This traditional model limits student engagement and weakens the connection between theoretical knowledge and musicianship. As national education reforms increasingly advocate for active, student-centered pedagogy [5], rethinking instructional approaches has become imperative. While research in Western contexts has demonstrated the potential of flipped learning to enhance engagement and performance in music education—particularly in areas such as music history and aural training [6,7]—such studies often lack contextual adaptation for non-Western systems. In Anglophone settings, factors such as digital readiness and learner autonomy are generally assumed [8,9], which may not hold in Chinese classrooms shaped by teacher-centered traditions. Addressing these gaps, the present study seeks not merely to replicate international models, but to develop a culturally responsive, theory-informed flipped classroom module tailored to the pedagogical realities of music theory instruction in China.

## 1.1. Adoption and limitations of flipped classroom in music education

The flipped classroom (FC) model has garnered increasing attention for its capacity to enhance student engagement and promote active learning by shifting core content delivery to the pre-class phase and repurposing classroom time for collaborative, inquiry-based activities [10,11]. While extensively adopted in STEM disciplines with positive effects on learning autonomy and achievement [12], its application in music education remains limited and often lacks theoretical and structural coherence.

Existing FC implementations in music theory frequently overlook the alignment between instructional design, domain-specific cognitive demands, and pedagogical theory. Without an integrated framework, students may experience fragmented learning, cognitive overload, or superficial engagement. Moreover, many studies rely on anecdotal outcomes or descriptive accounts, with few employing rigorous comparative methods or incorporating discipline-informed design principles.

Flipped classroom models developed primarily in Western educational contexts often emphasize learner autonomy and instructional flexibility. However, they seldom account for contextual constraints prevalent in non-Western education systems—such as rigid curricula, uneven technological access, and teacher-centered pedagogical traditions. To address these limitations, the present study integrates course design theory into a flipped instructional sequence specifically tailored to the cognitive and pedagogical characteristics of undergraduate music theory education in China [13,14].

## 1.2. Designing a discipline-specific flipped classroom module

To address the limitations of conventional music theory instruction in Chinese higher education, this study proposes a Flipped Classroom Module (FCM) specifically tailored to the cognitive and pedagogical demands of the discipline. The FCM is structured around a three-phase instructional model—pre-class preparation, in-class engagement, and post-class reinforcement—intended to support a full-cycle learning process. In the pre-class phase, students interact with multimedia content and structured tasks designed to cultivate cognitive readiness and autonomous learning

habits. During in-class sessions, collaborative strategies such as peer analysis, dialogic discussion, and musical problem-solving are used to deepen understanding. Post-class activities include reflection, formative feedback, and extended tasks aimed at consolidating knowledge and fostering transfer to real-world musical contexts.

The pedagogical design of the FCM is grounded in three interrelated theoretical frameworks. Self-Directed Learning (SDL) provides the foundation for pre-class autonomy, emphasizing learner agency and metacognitive regulation during independent engagement with content [15,16]. Course Development Theory supports the alignment of instructional goals, methods, and assessments across the three phases, thereby addressing the fragmentation often found in unsystematic flipped models [17]. Collaborative Learning Theory informs the structure of classroom interaction, positing that musical understanding—particularly in abstract areas such as harmony and analysis—is best constructed through social reasoning and peer-based inquiry [18].

The FCM is implemented through a design and development research (DDR) methodology that combines iterative instructional design with empirical validation. Pre- and post-test comparisons between an experimental group and a control group are used to assess learning gains in both cognitive and applied domains. To date, no existing flipped classroom model has systematically integrated these three theoretical perspectives within a domain-specific, multi-phase instructional structure for music theory. This study therefore not only responds to calls for pedagogical reform in Chinese higher education [5] but also contributes to the international discourse by offering a replicable, evidence-based framework for innovation in music pedagogy.

## 2. Literature review

### 2.1. The evolution and pedagogical impact of flipped learning across disciplines

Over the past decade, flipped learning has transitioned from an instructional innovation to a widely adopted pedagogical approach across disciplines [19,20]. While early adoption was most prominent in STEM fields, more recent research has expanded into arts and humanities, including language education, engineering, and health sciences [21–24]. Across contexts, empirical findings consistently demonstrate that well-designed flipped classrooms foster deeper learning, sustained engagement, and greater learner autonomy [25,26].

At its core, flipped pedagogy restructures the learning process by shifting knowledge transmission to pre-class environments—typically via digital materials—while repurposing classroom time for collaborative, inquiry-based tasks. This inversion encourages students to assume active cognitive roles and facilitates higher-order reasoning. Nevertheless, while the pedagogical model shows broad promise, its disciplinary applications remain uneven. Particularly in music education, where abstract conceptual knowledge must be integrated with performance-based reasoning, the theoretical and design coherence of flipped models is rarely examined [7,27].

### 2.2. Challenges and gaps in flipped learning within music education

Despite growing interest in flipped instruction in the arts, research within music education remains fragmented and methodologically limited. Early studies report positive outcomes related to student motivation and classroom participation [6], yet few of these initiatives are guided by formal instructional design frameworks or informed by theories of learning [8,28]. In the context of music theory, where cognitive abstraction and symbolic reasoning are central, many flipped implementations rely on generic video content and lack mechanisms for integrating analytical thinking with embodied practice.

A key issue is the disconnect between instructional phases. Pre-class activities often emphasize passive content consumption, while in-class engagement lacks scaffolding or alignment with disciplinary goals. Additionally, few models explicitly address post-class reinforcement, limiting opportunities for consolidation and skill transfer. As a result, learning experiences become fragmented, and students may fail to make meaningful connections between theory and musical application.

Moreover, most studies rely on student perceptions or descriptive reporting, with minimal use of validated assessments or control comparisons. Longitudinal impacts are rarely tracked, and few interventions are designed with replicability in mind. Although some researchers call for design-based approaches, methodologies such as Design and Development Research (DDR) remain underutilized in this domain, particularly in the performing arts, where iterative refinement could bridge theoretical ideals and practical classroom constraints.

### 2.3. Theoretical and design foundations for flipped music instruction

Addressing these limitations requires an integrated design approach grounded in relevant learning theories. Three theoretical frameworks—Self-Directed Learning (SDL), Course Development Theory, and Collaborative Learning Theory—offer complementary strengths for structuring flipped instruction in music theory. SDL emphasizes learner autonomy and metacognitive engagement, which are crucial during pre-class preparation [29]. Course Development Theory provides systematic alignment between objectives, content, and assessment across instructional phases, mitigating the incoherence often seen in ad hoc designs [30]. Meanwhile, Collaborative Learning Theory underscores the role of dialogic interaction and co-construction of knowledge during in-class engagement, making it especially relevant to music-theoretical analysis and interpretation [31].

Although these frameworks have been applied separately in educational research, their combined operationalization in music education—especially in the flipped format—remains rare. There is little empirical evidence of how they can be integrated into a coherent, discipline-specific instructional model or how such a model might be validated through systematic research.

The present study LMS (Learning Management System) responds to this gap by designing and evaluating a flipped classroom module for undergraduate music theory, explicitly grounded in these three theoretical strands. Unlike prior work that treats flipped learning as a fixed format, this study adopts a developmental perspective informed by DDR. As summarized in (Table 1), each phase of instruction—pre-class, in-class, and post-class—is aligned with a distinct theoretical function. This design not only enhances pedagogical coherence but also enables empirical investigation of its efficacy in a Chinese higher education context. In doing so, the study contributes to ongoing efforts to modernize music pedagogy through structured, theory-driven, and context-responsive instructional innovation.

## 3. Methods

### 3.1. Research hypotheses

In line with literature review results, and given that this study employed a quasi-experimental design to compare the Flipped Classroom Module (FCM) with traditional teaching methods, the following research hypotheses were formulated:

H1: there is a positive and significant difference on Musical Basic Knowledge among students in pre-test and post-test in both experiment and control group.

**Table 1. FCM Module Learning Theory Integration.**

| Instructional Phase | Theoretical Foundation | Pedagogical Implementation Example |
|---|---|---|
| Pre-class Preparation | Self-Directed Learning (SDL) | Students watch video lectures, complete quizzes, and review core concepts independently via the LMS to foster autonomy and metacognitive control. |
| In-class Collaboration | Collaborative Learning Theory | Small-group musical analysis, peer discussion, and interpretive tasks encourage co-construction of music-theoretical knowledge. |
| Post-class Reinforcement | Course Development Theory | Reflective assignments, formative feedback, and extension tasks ensure alignment with learning objectives and scaffold long-term retention |

H2: there is a positive and significant difference on Musical Competence among students in pre-test and post-test in both experiment and control group.

H3: there is a positive and significant difference on Musical Basic Knowledge between students in control and experiment group in pre-test and post-test.

H4: there is a positive and significant difference on Musical Competence between students in control and experiment group in pre-test and post-test.

To test these hypotheses, independent-sample t-tests were conducted to compare the post-test outcomes between the experimental and control groups (H1, H2), while paired-sample t-tests were employed to examine within-group improvements from pre-test to post-test (H3, H4).

### 3.2. Ethical concerns

This study was carefully designed to ensure that all participation by undergraduates was voluntary, non-invasive, and conducted with full respect for privacy and confidentiality. Data collection focused on students' general learning experiences, preferences, and perceptions regarding music theory instruction, without involving any sensitive or personally identifiable information. The potential benefits of the study—for both educational research and the student participants—include valuable insights into instructional design, evidence-based improvements to pedagogical practices, and a greater understanding of learner-centered strategies in higher music education. These benefits were considered to outweigh the minimal risks involved. Undergraduates were prospectively recruited from a university in Guangxi, China, between March 1, 2024, and April 15, 2024. Informed consent was obtained prior to participation. All participants were clearly informed about the purpose and procedures of the study, their voluntary involvement, and their right to withdraw at any time without penalty. Verbal consent was initially obtained from participants to confirm their willingness to participate, after which written consent was secured as a formal record (see supporting documents). All data—including test scores and survey responses—were collected and processed anonymously. The authors did not have access to any information that could identify individual participants during or after data collection. Data were accessed for research purposes on April 20, 2024. The study was approved by the Ethics Committee for Research Involving Human Subjects at Universiti Kebangsaan Malaysia (RECUKM), under reference number: JEP-2025–650 (see supporting document [3.0–3.7]), ensuring ethical compliance and participant confidentiality. Additional information regarding the ethical, cultural, and scientific considerations specific to inclusivity in global research is included in the supporting document (document [5.1]).

### 3.3. Course structure

The Flipped Classroom Module (FCM) was structured into a 12-week instructional program, comprising three integrated phases: pre-class, in-class, and post-class learning activities. Each week included one cycle of these phases. In the pre-class phase, students engaged in self-directed learning through short video lectures, readings, and online quizzes delivered via the Chaoxiang Learning Management System (LMS). The in-class phase emphasized active learning strategies, such as group discussions, collaborative problem-solving, and student presentations, facilitated by the instructor. The post-class phase involved reflective assignments, formative assessments, and individualized feedback, reinforced through mobile applications and online forums. The curriculum content aligned with national music theory standards and covered topics such as rhythm, melody, harmony, and musical forms. Both experimental and control groups were taught by the same instructor, using the same course content and assessments, to ensure instructional equivalence (Table 2) The instructor was a university lecturer with over ten years of experience teaching undergraduate music theory courses and was responsible for implementing the flipped classroom module..

**Table 2. Module Application.**

| Implementation phase | Duration |
|---|---|
| Enrolment guidance workshop | 4 hours |
| Pre-test | 1 hour and 30 minutes (n = 30 flipped classroom, n = 30 traditional teaching) |
| Intervention | A total of nine "pre-class" sessions (each lasting 45 minutes to 1 hours) is conducted via the Chaoxiang Learning Management System using mobile phones, tablets, or computers. Nine "in-class" sessions (each lasting 2–3 hours) are held through face-to-face meetings. Following these, nine "post-class" sessions (each lasting 45 minutes to 1 hour) are completed using the app for review purposes. |
| Post-test | 1 hour and 30 minutes (n = 30 flipped classroom, n = 30 traditional teaching) |

### 3.4. Experimental design process

A total of five classes of first-year music performance students were available for the course. Among them, two intact classes were selected to participate in the quasi-experimental study, with 30 students in each class, resulting in a total of 60 participants. These two classes were chosen because they had equal class sizes, while the other three classes had uneven numbers of students, which would have limited the comparability between groups. Using intact classes maintained natural learning conditions and avoided disruptions to normal teaching schedules. Both selected
classes were taught by the same instructor, followed the same syllabus, and used identical materials and assessments, ensuring equivalent learning conditions for both groups except for the instructional approach. The intervention consisted of three phases—pre-class learning, in-class interaction, and post-class reinforcement—aimed at restructuring the conventional instructional sequence through a technology-supported blended learning model. While instructional content, teaching staff, and assessment criteria were identical for both the experimental group (flipped classroom, n = 30) and the control group (traditional classroom, n = 30), the instructional mode was the sole differentiating factor. The experimental procedure is presented in (Fig 1).

### 3.5. Differences in implementation between the FCM experimental group and the control group

This study examined the differences between the FCM module under the flipped classroom model and the traditional instructional approach, focusing on variations in teaching structure, integration of technology, and educational objectives. The experimental group (FCM) employed a three-phase instructional model—pre-class self-directed learning via LMS,

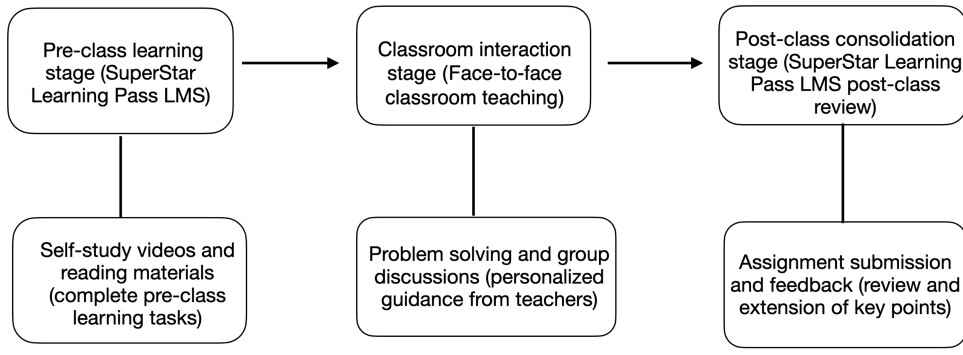

**Fig 1. Experimental design process.**

**Fig 2. Specific implementation differences between the experimental group and the control group.**

in-class collaborative engagement through discussion and group activities, and post-class personalized online feedback—emphasizing student-centered instruction and the development of higher-order cognitive skills. By contrast, the control group (traditional instruction) followed a conventional sequence of teacher-led lectures, standardized practice, and paper-based assignments, prioritizing systematic knowledge delivery but offering less timely feedback. As (Fig 2) resented, the findings underscore the potential of the FCM to enhance classroom time management, leverage digital tools, and support the cultivation of core competencies, offering theoretical insights for the digital transformation of education.

### 3.6. Course content and instructional platform

The course covered fundamental topics in music theory, including rhythm, pitch, scale construction, harmony, and chord progressions. Instructional materials and activities were aligned with national curriculum standards and delivered through a Moodle-based LMS. The platform included four key components: (1) course overview and syllabus, (2) weekly instructional materials (videos, lecture slides, exercises), (3) discussion forums, and (4) assignment submission and grading. Students accessed the LMS through computers, tablets, or smartphones, and were encouraged to review video lessons and practice exercises outside of classroom hours. Peer-to-peer discussion and Q&A threads within the LMS facilitated additional support.

### 3.7. Measurement

To evaluate the instructional effectiveness of the Flipped Classroom Module (FCM), this study utilized both cognitive and practical assessment tools. Students' learning outcomes were measured through pre-, and post-tests administered at the beginning and end of the 12-week course. These assessments were designed and validated by music education experts to ensure content alignment with course objectives and national standards in music theory education.

The cognitive knowledge test assessed students' understanding of core theoretical concepts, including scales, intervals, rhythm patterns, and harmonic functions. The musical element and skills test focused on the application of these concepts in practical contexts, such as melodic dictation, rhythmic transcription, and harmonic analysis.

Each test consisted of both multiple-choice and short-answer items, with scores standardized on a 25-point scale for comparability. In addition, students' subjective learning experiences and satisfaction were captured through a

post-intervention survey using a 5-point Likert scale, evaluating perceived engagement, clarity of instruction, self-efficacy, and the usefulness of the flipped format.

To ensure statistical validity, data were subjected to normality tests (Shapiro-Wilk and Kolmogorov–Smirnov), followed by paired-sample and independent-sample t-tests to compare intra-group and inter-group score differences. These analyses were used to determine whether observed changes were statistically significant and attributable to the instructional intervention.

### 3.8. Learning management system design

The instructional delivery for the experimental group was facilitated through a customized Moodle-based Learning Management System (LMS), specifically designed to support the pedagogical structure of the flipped classroom model. The LMS integrated four key functions into a cohesive digital platform: it provided students with an overview of the course structure, including weekly schedules, learning objectives, grading criteria, and administrative announcements. All instructional materials—such as pre-class video lectures (approximately 45 minutes to 1 hour per week), downloadable readings, weekly quizzes, and prompts for in-class activities—were made available through the system, allowing students to engage in self-paced learning prior to classroom sessions. The LMS also featured an interactive discussion forum that enabled students to pose questions, exchange ideas, and collaborate on assignments. In addition to LMS-based discussions, the instructor moderated group conversations on WeChat to provide immediate clarification and feedback, thereby extending instructional support beyond the classroom. Furthermore, the LMS incorporated mechanisms for assignment submission, progress tracking, and individualized feedback, with formative assessments embedded to monitor and guide students' learning throughout the course. Students could access the platform via desktop computers, tablets, or smartphones, and received automated push notifications regarding upcoming tasks and deadlines. To evaluate engagement, system usage data—including login frequency and video completion rates—were monitored and cross-referenced with student performance outcomes.

## 4. Results

### 4.1. Normality test (students' cognitive abilities) analysis

This study involved 60 undergraduate music students who were divided into two groups using a quasi-experimental design: the traditional classroom group (n = 30) and the flipped classroom (FCM) group (n = 30). To ensure the appropriateness of subsequent parametric tests, the study conducted a comprehensive normality assessment on the pre-test and post-test data for both groups, employing both descriptive statistics and inferential normality tests (Table 3).

From the descriptive statistics, the mean values for student cognitive abilities across the four datasets ranged from 13.90 to 16.43, indicating comparable performance levels between groups prior to the intervention and moderate

**Table 3. Results of normality tests (students' cognitive abilities).**

| Name | Sample size | Mean | Standard deviation | Skew-ness | Kur-tosis | Kolmogorov-Smirnov. inspection | | Shapiro-Wilk inspection | |
|---|---|---|---|---|---|---|---|---|---|
| | | | | | | Statistical value $D$ | $p$ | Statistical value $W$ | $p$ |
| Student Cognitive Ability Traditional Classroom – Pre-test | 30 | 13.90 | 4.18 | 0.02 | −1.15 | 0.13 | 0.20 | 0.95 | 0.20 |
| Student Cognitive Ability Traditional Classroom – Post-test | 30 | 14.13 | 3.95 | 0.16 | −0.68 | 0.15 | 0.08 | 0.94 | 0.08 |
| Student Cognitive Ability FCM Module – Pre-test | 30 | 13.73 | 3.94 | 0.09 | −1.00 | 0.12 | 0.20 | 0.95 | 0.15 |
| Student Cognitive Ability FCM Module – Post-test | 30 | 16.43 | 3.91 | −0.24 | −0.60 | 0.12 | 0.20 | 0.94 | 0.07 |

* $p < 0.05$ ** $p < 0.01$

improvement afterward, particularly in the FCM group. Specifically, the traditional classroom group scored a mean of 13.90 (SD = 4.18) in the pre-test and 14.13 (SD = 3.95) in the post-test, showing minimal change. The FCM group showed an increase from a mean of 13.73 (SD = 3.94) in the pre-test to 16.43 (SD = 3.91) in the post-test, reflecting a noticeable gain as illustrated in (Fig 3), the post-test scores of the FCM group show not only an upward shift in central tendency but also a narrower interquartile range, indicating increased consistency among learners.

The standard deviations across all datasets ranged from 3.91 to 4.18, suggesting a relatively consistent dispersion of scores within both groups. This uniformity enhances the reliability of subsequent comparisons and indicates that no extreme variability or clustering affected the score distributions.

The skewness values for the four datasets—traditional pre-test (0.02), traditional post-test (0.16), FCM pre-test (0.09), and FCM post-test (–0.24)—all fell within the conventional ±1 threshold, indicating near-symmetric distributions. Similarly, the kurtosis values ranged from –1.15 to –0.60, all within the acceptable range of ±2, implying no excessive peaked Ness or flatness in the score distributions (Kline, 2015).

(Fig 4) presents the trajectory of average cognitive scores across the pre- and post-tests. The FCM group demonstrated a more pronounced upward trend, whereas the traditional group remained largely static. Together, these descriptive indicators support the assumption of approximately normal distributions for all datasets. In terms of inferential normality testing, this study adhered to the methodological recommendations of Mooi and Sarstedt (2011) and Field (2018), giving priority to the Shapiro-Wilk test—recognized for its robustness in small to medium sample sizes ($n < 50$)—and used the Kolmogorov-Smirnov (K-S) test as a secondary measure.

For the traditional classroom group, the pre-test showed no significant deviation from normality (Shapiro-Wilk: W = 0.95, p = 0.20; K-S: D = 0.13, p = 0.20). The post-test showed borderline normality (Shapiro-Wilk: W = 0.93, p = 0.05; K-S: D = 0.17, p = 0.03) but given the sensitivity of the K-S test and the borderline p-value from the Shapiro-Wilk test, the distribution was still considered approximately normal. This is further supported by acceptable skewness (0.16), kurtosis (–0.68), and the absence of outliers.

For the FCM group, both the pre-test (Shapiro-Wilk: W = 0.95, p = 0.15; K-S: D = 0.12, p = 0.20) and post-test (Shapiro-Wilk: W = 0.94, p = 0.08; K-S: D = 0.13, p = 0.18) met the criteria for normality. Descriptive metrics—such as the post-test skewness (–0.24) and kurtosis (–0.60)—further confirmed that the data conformed to normal distribution assumptions.

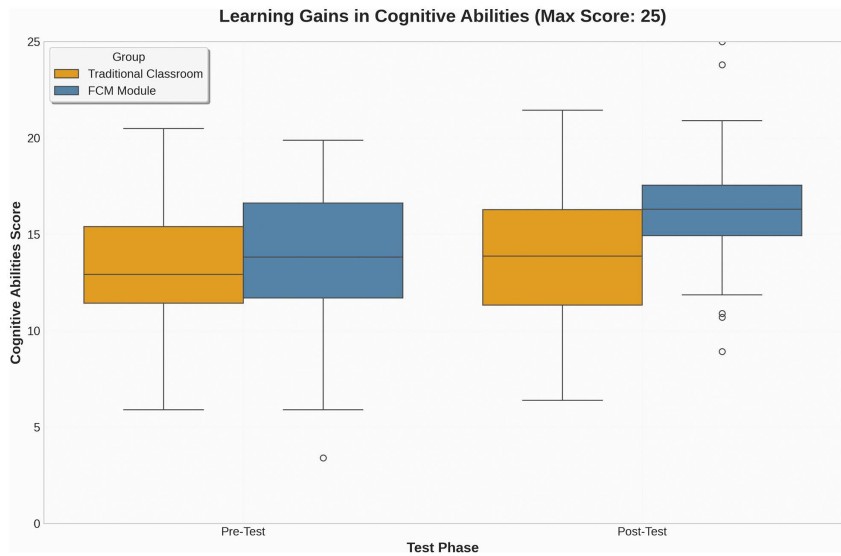

**Fig 3. Pre- and Post-Test Cognitive Scores.**

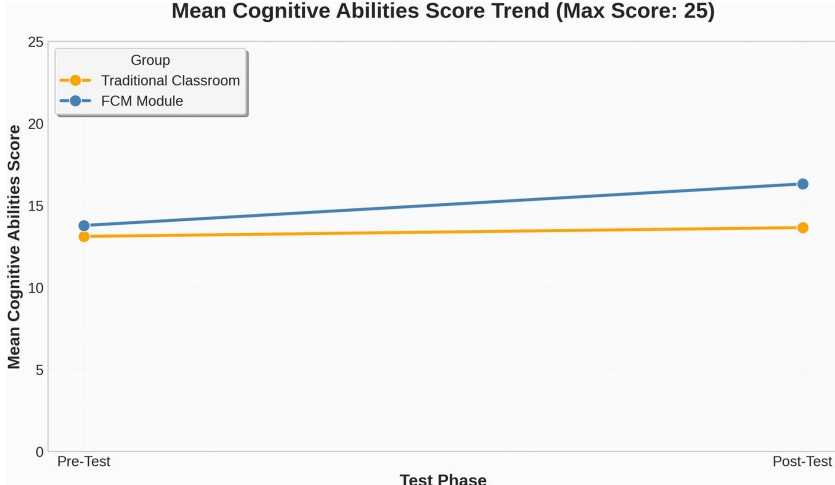

**Fig 4. Cognitive Learning Gain Trends.**

Taken together, the findings from both descriptive and inferential analyses confirm that none of the four datasets demonstrated significant violations of the normality assumption. Although the traditional classroom post-test result approached the threshold of statistical significance, the Shapiro-Wilk test remains the preferred benchmark, and descriptive metrics showed acceptable distribution characteristics.

Furthermore, in alignment with the Central Limit Theorem (Field, 2018), parametric procedures such as the t-test are robust to moderate deviations from normality when the sample size is approximately n ≥ 30, as in the present study. Therefore, these datasets provide a valid foundation for subsequent parametric analyses of cognitive learning outcomes.

## 4.2. Normality test (musical elements and skills)

The results of normality tests for musical elements and skills in both the traditional classroom and flipped classroom (FCM) groups. The analysis includes descriptive statistics (mean, standard deviation, skewness, and kurtosis) as well as inferential indicators (Shapiro-Wilk and Kolmogorov-Smirnov tests) (Table 4).

From the descriptive statistics, the pre-test means for the traditional classroom (M = 16.30, SD = 4.36) and the FCM group (M = 17.57, SD = 2.19) indicate relatively comparable baseline performance, although the FCM group exhibited slightly higher average scores and smaller variability. In the post-test phase, the FCM group's mean score increased to

**Table 4. Results of normality tests (musical elements and skills).**

| Name | Sample size | Mean | Standard deviation | Skew-ness | Kur-tosis | Kolmogorov-mirnov inspection | | Shapiro-Wilk inspection | |
|---|---|---|---|---|---|---|---|---|---|
| | | | | | | Statistical value $D$ | $p$ | Statistical value $W$ | $p$ |
| Music Elements Skills Traditional Classroom – Pre-test | 30 | 16.30 | 4.36 | −0.72 | 0.06 | 0.15 | 0.09 | 0.93 | 0.05 |
| Music Elements Skills Traditional Classroom – Post-test | 30 | 16.90 | 4.46 | −0.51 | −0.18 | 0.16 | 0.04 | 0.97 | 0.42 |
| Music Elements Skills FCM Module – Pre-test | 30 | 17.57 | 2.19 | −0.29 | −0.71 | 0.18 | 0.02 | 0.94 | 0.07 |
| Music Elements Skills FCM Module – Post-test | 30 | 19.70 | 2.12 | −0.25 | −0.47 | 0.14 | 0.16 | 0.93 | 0.06 |

* $p < 0.05$ ** $p < 0.01$

19.70 (SD = 2.12), compared to 16.90 (SD = 4.46) in the traditional classroom. Notably, the standard deviation in the FCM group decreased slightly, suggesting that students' performance became more consistent, while variability in the traditional group remained higher. (Fig 5) provides a visual representation of this distributional change. The boxplot reveals not only an upward shift in median scores for the FCM group but also a narrower interquartile range in the post-test, indicating greater uniformity and less dispersion in student performance. These results suggest that the FCM may have enhanced not only learning outcomes but also reduced performance disparity among learners. As further illustrated in (Fig 6), the FCM group demonstrated a more linear and consistent upward trajectory in skill-based scores, whereas the traditional classroom group exhibited only marginal improvement with continued variability.

In terms of distributional shape, all four datasets showed mild negative skewness, ranging from –0.72 to –0.25. Specifically, skewness was –0.72 (traditional pre-test), –0.51 (traditional post-test), –0.29 (FCM pre-test), and –0.25 (FCM post-test). These values fall within the acceptable ±1 range, suggesting moderate symmetry. Likewise, kurtosis values ranged from –0.71 to +0.06, all well within the ± 2 criterion (Kline, 2015), indicating no significant issues of peaked Ness or flatness in any group. These characteristics are generally consistent with the assumption of normality.

Regarding inferential normality tests, the study followed the standard procedure of prioritizing the Shapiro-Wilk (S-W) test for small to medium-sized samples (n < 50), while using the Kolmogorov-Smirnov (K-S) test as a secondary reference (Mooi & Sarstedt, 2011; Field, 2024).

For the traditional classroom group, the pre-test showed acceptable normality in both tests: Shapiro-Wilk (W = 0.93, p = 0.05) and Kolmogorov-Smirnov (D = 0.15, p = 0.09). Although the S-W p-value was borderline, the skewness (–0.72) and kurtosis (0.06) remained within acceptable limits, supporting the decision to proceed with parametric tests. The post-test results also supported normality: Shapiro-Wilk (W = 0.97, p = 0.42) and K-S (D = 0.16, p = 0.04). While the K-S test was statistically significant, the S-W result and descriptive indicators (skewness = –0.51; kurtosis = –0.18) did not signal a substantial deviation. Thus, consistent with Field (2024), the more reliable S-W test result was prioritized.

For the FCM group, both pre- and post-test results also indicated approximate normality. The pre-test (W = 0.94, p = 0.07; D = 0.18, p = 0.02) suggested slight deviation in the K-S test, but not in the S-W test, and descriptive statistics

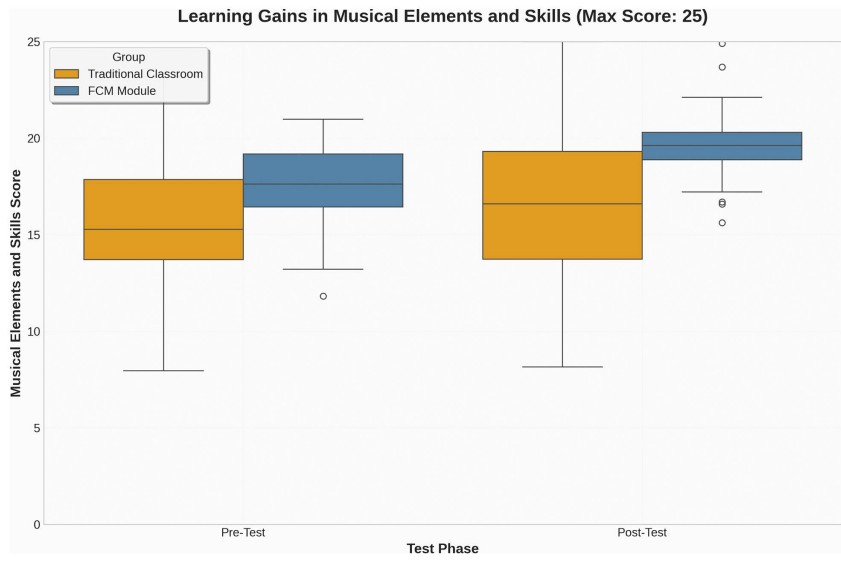

**Fig 5. Pre- and Post-Test Musical Skills Scores.**

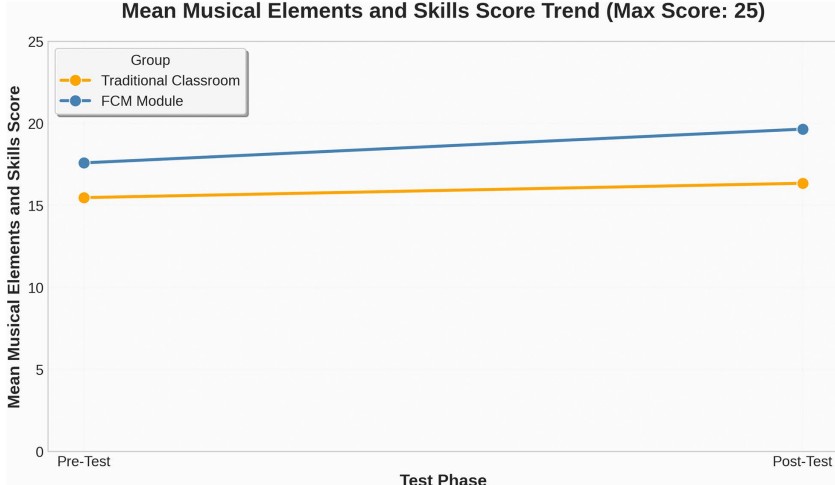

**Fig 6. Musical Skill Learning Gain Trends.**

(skewness = −0.29; kurtosis = −0.71) remained well within conventional thresholds. The post-test confirmed normality: S-W (W = 0.93, p = 0.06) and K-S (D = 0.14, p = 0.16), with skewness (−0.25) and kurtosis (−0.47) also within acceptable bounds.

Taken together, none of the four datasets violated the assumption of normality in a statistically or practically significant manner. Even in cases where the Kolmogorov-Smirnov test returned marginal p-values (e.g., FCM pre-test), the Shapiro-Wilk test results and descriptive distribution indices (skewness, kurtosis, SD) suggest that the data were sufficiently normal for parametric analysis, particularly given the robustness of t-tests to moderate deviations in normality for sample sizes of n ≥ 30 (Field, 2024).Therefore, the datasets for musical elements and skills meet the essential assumptions required for the application of paired-sample t-tests and independent-sample t-tests in subsequent analysis stages.

### 4.3. Comprehensive analysis of independent and paired sample t-Tests

To examine the effects of the flipped classroom module (FCM) on student learning outcomes, independent samples t-tests were conducted to compare the traditional classroom group and the FCM group across two key variables—cognitive abilities and musical element skills—at both the pre-test and post-test stages. As shown in (Table 5), the analysis included four statistical comparisons to evaluate the instructional effects of the FCM.

In the pre-test stage, no statistically significant differences were found between the two groups in either cognitive ability or musical element skills. Specifically, for cognitive ability, the traditional classroom group achieved a mean score of 13.90 (SD = 4.18), while the FCM group scored 13.73 (SD = 3.94), yielding a non-significant result, t (58) = 0.16, p = .87. Similarly,

**Table 5. Independent samples t-test analysis results.**

| Name | Group (mean ± standard deviation) | | $t$ | $p$ |
|---|---|---|---|---|
| | Traditional classroom ($n$ = 30) | Flipped classroom ($n$ = 30) | | |
| Student Cognitive Abilities – Pre-test | 13.90 ± 4.18 | 13.73 ± 3.94 | 0.16 | 0.87 |
| Student Cognitive Abilities – Post-test | 14.13 ± 3.95 | 16.43 ± 3.91 | −2.27 | 0.03 |
| Music Element Skills – Pre-test | 16.30 ± 4.36 | 17.57 ± 2.19 | −1.42 | 0.16 |
| Music Element Skills – Post-test | 16.90 ± 4.46 | 19.70 ± 2.12 | −3.11 | 0.00 |

\* $p$ < 0.05 \*\* $p$ < 0.01

in musical element skills, the traditional group scored 16.30 (SD = 4.36), and the FCM group scored slightly higher at 17.57 (SD = 2.19), though the difference was not statistically significant, t (58) = –1.42, p = .16. These findings confirm the baseline equivalence between the two groups prior to intervention, thereby strengthening the internal validity of the subsequent comparative analysis.

Following the intervention, significant differences emerged in both outcome variables, favoring the FCM group. In terms of cognitive ability, the FCM group demonstrated a notable improvement, with a mean score of 16.43 (SD = 3.91), compared to 14.13 (SD = 3.95) in the traditional group. The observed difference reached statistical significance, t (58) = –2.27, p = .03, suggesting that the flipped classroom approach had a positive impact on students' cognitive performance. A similar trend was observed in musical element skills, where the FCM group achieved a substantially higher post-test score of 19.70 (SD = 2.12), compared to 16.90 (SD = 4.46) in the traditional classroom group. This difference was highly significant, t (58) = –3.11, p < .01, indicating that the FCM model was not only effective in enhancing students' musical proficiency but also contributed to reducing performance variability among learners, as evidenced by the smaller standard deviation in the FCM group.

In summary, the results of the four independent samples t-tests demonstrate that while the two groups were comparable at the pre-test stage, the FCM group outperformed the traditional group in both cognitive and practical musical domains after the intervention. These findings provide empirical support for the effectiveness of flipped classroom pedagogy in the context of undergraduate music theory instruction. By controlling for content, instructor, and assessment conditions across groups, the study isolates the instructional model as the primary variable influencing learning outcomes. Therefore, the evidence highlights the pedagogical value of the FCM in promoting higher-order thinking, learner autonomy, and enhanced academic achievement in both theoretical understanding and applied musical skill.

The consistency in pre-test scores underscores the robustness of the experimental design, while the post-test results underscore the efficacy of the FCM in fostering measurable improvements in student performance. Future research could explore longitudinal effects and the scalability of this approach across diverse educational settings.

### 4.4. Paired-sample t-Test on cognitive knowledge (learning gains)

To further investigate the impact of the flipped classroom module (FCM) on students' cognitive development within each instructional group, paired-sample t-tests were conducted to assess the differences between pre-test and post-test scores for both the traditional classroom group and the FCM group. This within-group comparison complements the between-group findings reported earlier and helps isolate the learning gains directly attributable to each instructional approach. The results are summarized in (Table 6).

In the traditional classroom group, the pre-test mean score was 13.90 (SD = 4.18), while the post-test mean score showed a marginal increase to 14.13 (SD = 3.95). The difference between the two time points was minimal (mean difference = −0.23), and the paired-sample t-test revealed no statistically significant change, t (29) = −0.46, p = .65. This

**Table 6. Results of t-tests for cognitive knowledge (learning gains) matching samples.**

| Name | Grouping | Pairing (mean ± standard deviation) | | Difference (Pair 1 – Pair 2) | t | p |
|---|---|---|---|---|---|---|
| | | Before the experiment | After the experiment | | | |
| Traditional Classroom Student Cognitive Abilities – Pre-test & Student Cognitive Abilities – Post-test | Control group | 13.90 ± 4.18 | 14.13 ± 3.95 | −0.23 | −0.46 | 0.65 |
| Flipped Classroom FCM Module Student Cognitive Abilities – Pre-test & Student Cognitive Abilities – Post-test | Experimental group | 13.73 ± 3.94 | 16.43 ± 3.91 | −2.70 | −3.94 | 0.00 |

\* p < 0.05 \*\* p < 0.01

indicates that students receiving conventional instruction did not exhibit meaningful improvement in cognitive performance over the course of the intervention.

By contrast, the FCM group demonstrated a substantial improvement in cognitive outcomes. The pre-test mean score was 13.73 (SD = 3.94), which increased significantly to 16.43 (SD = 3.91) in the post-test. The mean difference of −2.70 (post-test minus pre-test) was statistically significant, with t (29) = −3.94, p < .001. These results provide robust evidence that the flipped classroom model effectively enhanced students' cognitive learning within the duration of the study.

Taken together, the findings from the paired-sample t-tests reinforce the results of the independent samples t-tests. While both groups started at statistically equivalent baselines, only the FCM group achieved significant intra-group cognitive gains, confirming that the flipped classroom model not only yielded better comparative outcomes but also produced statistically meaningful progress within the group itself. This aligns with pedagogical theories emphasizing active engagement and pre-class preparation as key contributors to improved cognitive achievement in higher education contexts. The consistency of these results across both within-group and between-group analyses strengthens the validity of the conclusion that the FCM approach is more effective than traditional instruction for fostering cognitive development in music theory education.

## 4.5. Paired-sample t-Test on musical elements and skills

To complement the analysis of cognitive outcomes, a paired-sample t-test was conducted to examine whether students in each instructional group demonstrated significant improvement in musical element skills as a result of the intervention. This analysis offers a within-group comparison of students' practical musical competencies before and after the instructional treatment and serves to evaluate the impact of both teaching models on applied music learning. The results are presented in (Table 7).

For the traditional classroom group, the pre-test mean score was 16.30 (SD = 4.36), while the post-test mean was slightly higher at 16.90 (SD = 4.46). Despite this minor improvement (mean difference = −0.60), the paired-sample t-test did not yield a statistically significant result, t (29) = −1.58, p = .13. This indicates that the conventional instructional approach did not result in meaningful enhancement of students' musical skills over the course of the study.

In contrast, the FCM group showed a significant and consistent improvement. The pre-test mean score was 17.57 (SD = 2.19), which increased to 19.70 (SD = 2.12) after the intervention. The observed mean gain of −2.13 points was statistically significant, t (29) = −6.31, p < .001. This finding suggests that the flipped classroom model had a robust and positive impact on students' acquisition of practical music skills, potentially due to increased opportunities for active engagement, performance-based application, and individualized feedback throughout the instructional process.

These results echo the patterns observed in the cognitive domain, further supporting the flipped classroom's effectiveness in enhancing both theoretical understanding and practical musical abilities. Whereas the traditional approach

**Table 7. Paired-Sample t-Test Results on Musical Elements and Skills.**

| Name | Grouping | Pairing (mean ± standard deviation) | | Difference (Pair 1 − Pair 2) | t | p |
|---|---|---|---|---|---|---|
| | | Before the experiment | After the experiment | | | |
| Traditional Classroom Music Element Skills – Pre-test & Music Element Skills – Post-test | Control group | 16.3 ± 4.36 | 16.9 ± 4.46 | −0.6 | −1.58 | 0.13 |
| Flipped Classroom FCM Module Music Element Skills – Pre-test & Music Element Skills – Post-test | Experimental group | 17.57 ± 2.19 | 19.7 ± 2.12 | −2.13 | −6.31 | 0.00 |

\* p < 0.05 \*\* p < 0.01

appeared insufficient in driving significant within-group gains, the FCM intervention yielded substantial improvements across both learning dimensions. The alignment of results across independent and paired-sample tests reinforces the conclusion that the FCM model is pedagogically advantageous in the context of undergraduate music theory and musicianship training.

## 5. Discussion

### 5.1. Empirical gains from a theory-informed flipped classroom design

This study provides empirical support for the effectiveness of a theory-integrated Flipped Classroom Module (FCM) in enhancing both conceptual understanding and practical musicianship in undergraduate music theory education. The findings indicated significant improvements in both Student Cognitive Abilities and Music Element Skills from pre-test to post-test in the experimental group. Moreover, students who participated in the Flipped Classroom Module (FCM) achieved notably higher post-test results than those in the traditional classroom, demonstrating that the FCM effectively enhanced students' cognitive development and mastery of musical elements. These findings validate those four hypotheses that a systematically structured flipped model can achieve superior learning outcomes in domain-specific educational settings.

The results resonate with evidence from STEM and language education, where structured pre-class preparation and active in-class learning have been shown to improve engagement and knowledge retention [31,32]. In contrast to prior music education studies that relied on anecdotal insights or lacked methodological rigor [9,33], this study contributes controlled empirical evidence anchored in validated measures. Compared to descriptive reports such as [6], which examined engagement in a flipped composition course without experimental control, the present research demonstrates how theory-informed instructional sequencing can yield both measurable and replicable learning effects.

### 5.2. Bridging design gaps through integrated pedagogical frameworks

The integration of Self-Directed Learning, Course Development Theory, and Collaborative Learning Theory into the instructional sequence represents a critical innovation in this study. Each theoretical framework informs a distinct phase of the FCM: autonomous pre-class engagement, dialogic in-class interaction, and scaffolded post-class reflection. This alignment provides coherence across the learning cycle and ensures that instructional design responds to both cognitive demands and contextual realities.

The pre-class phase cultivates learner agency and metacognitive preparation, consistent with Knowles' (1975) model of adult learning. In-class activities grounded in social negotiation mirror Dillenbourg's (1999) concept of collaborative knowledge construction. The post-class phase, reinforced by LMS-based feedback and mobile interactions, reflects design-based principles from Reigeluth and Carr-Chellman (2009) that promote retention and reflective transfer.

Compared to earlier flipped music classroom studies, which often foreground technology use over theoretical integration [8,9], this research advances a more intentional model rooted in pedagogical logic. By situating the design within the sociocultural and institutional conditions of Chinese higher education, the study expands the flipped classroom paradigm beyond Anglophone contexts and demonstrates its adaptability to systems characterized by teacher-centered norms and exam-oriented curricula.

### 5.3. Implications for music pedagogy and curriculum reform

The FCM developed in this study offers an actionable framework for enhancing instructional practice in music theory. The learning management system functioned not only as a content repository but also as a dynamic platform for fostering learner autonomy, enabling peer collaboration, and facilitating instructor feedback. The integration of synchronous and asynchronous tools—such as WeChat and Moodle—enabled responsive and culturally compatible teaching strategies.

The dual emphasis on conceptual clarity and applied performance supports a pedagogical model that bridges theoretical abstraction with creative practice. This alignment is particularly valuable in music theory instruction, where students often struggle to connect analytical knowledge with expressive execution. The model encourages active learning without compromising curricular standards or disciplinary rigor.

These findings align with ongoing reforms in Chinese higher education that advocate for inquiry-driven, student-centered, and digitally supported pedagogy. The structure and scalability of the FCM allow for institutional adaptation across diverse conservatories and comprehensive universities, while remaining consistent with national curriculum guidelines.

### 5.4. Limitations and directions for future research

Several limitations should be acknowledged in interpreting the study's findings. The research was conducted at a single institution, which may affect external validity. Broader implementation across multiple institutions and regional contexts is necessary to assess generalizability. The relatively short 12-week duration also limited the ability to evaluate long-term retention or transfer of skills.

While learning outcomes were measured through validated assessments, other dimensions—such as student motivation, creativity, and musical self-efficacy—were not examined. These areas warrant exploration in future studies using mixed methods approaches. In addition, individual differences in prior training and technological readiness were not controlled, which may have influenced intervention efficacy. Stratified sampling and statistical control for covariates could strengthen future research designs.

### 5.5. Conclusion and recommendations

This study demonstrates that a theory-informed Flipped Classroom Module (FCM) can effectively enhance students' cognitive understanding and practical musicianship in undergraduate music theory instruction. Grounded in Self-Directed Learning, Course Development Theory, and Collaborative Learning Theory, the FCM established a coherent three-phase instructional structure—pre-class preparation, in-class collaboration, and post-class reinforcement—that significantly improved learning outcomes compared to traditional teaching. The results confirm that integrating pedagogical theory with technological tools can foster learner autonomy, interaction, and reflective engagement, thereby addressing long-standing limitations of teacher-centered instruction in Chinese higher education. The study contributes a replicable and culturally responsive model for music theory pedagogy and provides empirical evidence supporting the transformative potential of flipped learning in the arts. It is recommended that educators adopt theory-driven flipped frameworks in curriculum design and receive targeted training in digital pedagogy to ensure effective implementation. Future research should examine the long-term effects, scalability, and cross-disciplinary applications of such models to further advance innovation in higher music education.

## Supporting information

**S1 File. Pre test and post test questionnaire.**
(DOCX)

**S2 File. Inclusivity in global research document.**
(PDF)

## Author contributions

**Conceptualization:** QISEN ZHU, Khairul Azhar Jamaludin, Jinglong Li.

**Data curation:** QISEN ZHU, Khairul Azhar Jamaludin, Jinglong Li.

**Methodology:** QISEN ZHU, Jinglong Li.

**Project administration:** QISEN ZHU, Khairul Azhar Jamaludin.

**Supervision:** Khairul Azhar Jamaludin, Jinglong Li.

**Validation:** Jinglong Li.

**Writing – original draft:** QISEN ZHU.

**Writing – review & editing:** QISEN ZHU.

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
