## [Decision Letter · Decision Letter 0]

20 Oct 2025

Dear Dr. ZHU,

Kind regards,

Semiu Olawale Makinde, Ph.D.

Academic Editor

PLOS ONE

3. In the ethics statement in the Methods, you have specified that verbal consent was obtained. Please provide additional details regarding how this consent was documented and witnessed, and state whether this was approved by the IRB.

4. In the online submission form, you indicated that [Summary-level data underlying the findings are provided in the manuscript. Full raw datasets are available from the corresponding author upon reasonable request.].

Additional Editor Comments:

This is a well-conceived and competently written paper addressing a timely issue in music education—the adaptation of the flipped classroom model to non-Western higher education. The theoretical integration of self-directed learning, course development theory, and collaborative learning theory is commendable and clearly operationalised in the instructional design.

The methodology is sound, and results are convincingly analysed. I particularly appreciate the clarity of tables and the balance between quantitative rigour and pedagogical interpretation.

For improvement, please:

1. Review the manuscript for minor typographical and grammatical errors.

2. Ensure consistency in citation formatting (e.g., duplication of some references such as Bolden et al., 2021 and Ng et al., 2022).

3. Shorten the discussion slightly to avoid repetition of points already stated in the results.

4. Clarify briefly in the methodology how reliability or validity of the tests was established (mention expert review or pilot testing more explicitly).

These are minor revisions that will further strengthen the manuscript’s clarity and presentation. Overall, it is a strong contribution suitable for publication after minor corrections.

Reviewers' comments:

Reviewer's Responses to Questions

**Comments to the Author**

1. Is the manuscript technically sound, and do the data support the conclusions?

Reviewer #1: Yes

Reviewer #2: Yes

2. Has the statistical analysis been performed appropriately and rigorously?

Reviewer #1: Yes

Reviewer #2: Yes

3. Have the authors made all data underlying the findings in their manuscript fully available?

Reviewer #1: Yes

Reviewer #2: Yes

4. Is the manuscript presented in an intelligible fashion and written in standard English?

Reviewer #1: Yes

Reviewer #2: Yes

Reviewer #1: The title of the paper is apt, while the Abstract is very good and is within the maximum number of words. The entire body of the work is coherent and is focused on the title of the paper. The results and analysis were explanatory enough. The references are in line with the APA 7th edition. The little areas pointed out can be viewed in the manuscript for necessary attention. On a final note, I hereby recommend that the manuscript be considered for publication after attending to the minor corrections.

Reviewer #2: The manuscript is able to provide good theoretical framework for the research and establish the gap. However, there are some assertions that needs to be substantiated in the background and literature review section. The literature review section can include similar research conducted in STEM areas. Do State objectives and the research hypotheses tested in the study. the Also, in the methodology section, kindly provide the number of items for each subsection of the research instrument used and the psychometric properties.

**Do you want your identity to be public for this peer review?** For information about this choice, including consent withdrawal, please see our Privacy Policy

Reviewer #1: No

Reviewer #2: No

---

## [Author Response · Author response to Decision Letter 1]

3 Nov 2025

All comments have been carefully addressed in the revised manuscript. Detailed revisions are highlighted in the revised version and summarized in the response report �see attachments� :

1.Is there anything called “Western flipped Classroom Models” if yes, justify you claim with source(s)

Thank you for your comment. We have revised the phrase to “flipped classroom models developed primarily in Western educational contexts” for clarity and added supporting sources to justify this claim.(Page 2, chapter 1.1, paragraph 3.)

2.What’s the full meaning of LMS? Before using abbreviation there is a need to state the full meaning at first use. Then subsequently the abbreviation can be used.

Thank you for your comment. We have revised the manuscript to spell out the full term at its first mention as “Learning Management System (LMS)” and used the abbreviation in subsequent occurrences for clarity and consistency.(Page 5, chapter 2.3, paragraph 3.)

3.Undergraduates should be sufficient rather than saying “undergraduate students”

Thank you for your suggestion. The word “students” has been removed.(The whole manuscript (page 5 and 6).

4.Who is the instructor?

The instructor was a university lecturer with over ten years of experience teaching undergraduate music theory courses and was responsible for implementing the flipped classroom module.(Page 7, chapter 3.3)

5.What’s your justification for having Flipped classroom to be 30 and control group is also 30. I would have expected you to use intact class.

Thank you for your comment. In this study, two intact classes were selected from a total of five first-year music performance classes. Each class had 30 students, while the other three classes had uneven sizes, which would have limited comparability. Using these two intact classes ensured balanced group sizes, ecological validity, and natural learning conditions, with both groups taught by the same instructor under identical instructional settings except for the teaching approach.(Page 7)

6. Both groups consisted of two intact classes from the same academic year and music theory program to ensure ecological validity and minimize cross-group contamination.

Thank you for the comment. The text has been revised to clarify that two intact classes (30 students each) were selected from five available classes to ensure equal group sizes and ecological validity. Both classes were taught by the same instructor under identical conditions, differing only in instructional approach.(Page 7)

7.There is a need to state your formulated hypotheses after Literature Review.

Thank you for your comments, I added a new chapter 3.1 research hypotheses to state those four hypotheses after literature review.(Page 5-6, chapter 3.1)

8.Conclusion and Recommendations

Thank you for your valuable comment. The Conclusion and Recommendations section has been revised to provide a clear summary of the study’s key findings, theoretical grounding, and practical implications. The revised section now highlights how the Flipped Classroom Module (FCM) enhanced students’ cognitive understanding and practical musicianship, outlines its theoretical foundations, and offers actionable recommendations for educators and future research. This addition ensures that the section fully meets the reviewer’s expectations for clarity, coherence, and academic contribution.(P=20, chapter 5.5)

---

## [Editor Report · Decision Letter 1]

11 Nov 2025

Development of Flipped Classroom Module FCM for Music Theory Instruction: An Innovative Approach to Music Education

PONE-D-25-42105R1

Dear Dr. QISEN ZHU

We’re pleased to inform you that your manuscript has been judged scientifically suitable for publication and will be formally accepted for publication once it meets all outstanding technical requirements.

Kind regards,

Semiu Olawale Makinde, Ph.D.

Academic Editor

PLOS ONE
---

## [Editor Report · Acceptance letter]

PONE-D-25-42105R1

PLOS ONE

Dear Dr. ZHU,

I'm pleased to inform you that your manuscript has been deemed suitable for publication in PLOS ONE. Congratulations! Your manuscript is now being handed over to our production team.

Kind regards,

on behalf of

Dr. Semiu Olawale Makinde

Academic Editor

PLOS ONE